# Lab-Made Electronic Nose for Fast Detection of *Listeria monocytogenes* and *Bacillus cereus*

**DOI:** 10.3390/vetsci7010020

**Published:** 2020-02-09

**Authors:** Prima Febri Astantri, Wredha Sandhi Ardha Prakoso, Kuwat Triyana, Tri Untari, Claude Mona Airin, Pudji Astuti

**Affiliations:** 1Veterinary Science, Faculty of Veterinary Medicine, Gadjah Mada University, Yogyakarta 55281, Indonesia; prima.febri@gmail.com (P.F.A.); sandhi.soemarto@gmail.com (W.S.A.P.); 2Animal Quarantine, Agriculture Quarantine Agency, Ministry of Agriculture, Jakarta 12550, Indonesia; 3Department of Physics, Faculty of Mathematics and Natural Science, Gadjah Mada University, Yogyakarta 55281, Indonesia; 4Institute of Halal Industry and Systems (IHIS) Gadjah Mada University, Yogyakarta 55281, Indonesia; 5Department of Microbiology, Faculty of Veterinary Medicine, Gadjah Mada University, Yogyakarta 55281, Indonesia; t_untari@ugm.ac.id; 6Department of Physiology, Faculty of Veterinary Medicine, Gadjah Mada University, Yogyakarta 55281, Indonesia; monaairin@ugm.ac.id

**Keywords:** *Listeria monocytogenes*, *Bacillus cereus*, electronic nose, LDA, QDA, SVM

## Abstract

The aim of this study is to determine the performance of a lab-made electronic nose (e-nose) composed of an array of metal oxide semiconductor (MOS) gas sensors in the detection and differentiation of *Listeria monocytogenes* (*L. monocytogenes*) and *Bacillus cereus* (*B. cereus*) incubated in trypticsoy broth (TSB) media. Conventionally, the detection of *L. monocytogenes* and *B. cereus* is often performed by enzyme link immunosorbent assay (ELISA) and polymerase chain reaction (PCR). These techniques require trained operators and expert, expensive reagents and specific containment. In this study, three types of samples, namely, TSB media, *L. monocytogenes* (serotype 4b American Type Culture Collection (ATCC) 13792), and *B. cereus* (ATCC) 10876, were used for this experiment. Prior to measurement using the e-nose, each bacterium was inoculated in TSB at 1 × 10^3^–10^4^ CFU/mL, followed by incubation for 48 h. To evaluate the performance of the e-nose, the measured data were then analyzed with chemometric models, namely linear and quadratic discriminant analysis (LDA and QDA), and support vector machine (SVM). As a result, the e-nose coupled with SVM showeda high accuracy of 98% in discriminating between TSB media and *L. monocytogenes*, and between TSB media and *B. cereus*. It could be concluded that the lab-made e-nose is able to detect rapidly the presence of bacteria *L. monocytogenes* and *B. cereus* on TSB media. For the future, it could be used to identify the presence of *L. monocytogenes* or *B. cereus* contamination in the routine and fast assessment of food products in animal quarantine.

## 1. Introduction

Foodborne illness has become a critical issue in global public health to date. According to WHO (2015), almost 600 million disease cases caused by the consumption of contaminated food by pathogenic bacteria. *Salmonella sp., Escherichia coli* O157:H7, *Staphylococcus aureus, Listeria monocytogenes*, and *Bacillus cereus* are known pathogens in food [1,2]. Those bacteria are a threat to ready-to-eat foods. They survive in an unfavorable environment during food production and storage (i.e., low pH, low temperature, and high salt) [3].

*Listeria monocytogenes* causes the highest case of hospitalization (up to 91%) among other foodborne illnesses [4]. Listeriosis is infectious to humans and mammals, including the ruminant and monogastric animals. The clinical signs of listeriosis in humans include gastroenteritis, diarrhea, meningitis, bacteremia, and it causes encephalitis, septicemia, abortion, mastitis, and gastroenteritis in cows [5,6]. Member of genus *Listeria* is a non-spore bacterium, being anaerobic facultative, a small size, Gram-positive, and rod-shaped (0.5–4.0 µm diameter and 0.5–2.0 µm long). *Listeria monocytogenes* can contaminate a wide range of foods, including yogurt, cheese, meat, ham, smoked salmon, poultry, seafood and vegetable products [2,7].

*Bacillus cereus* is a facultative aerobic to anaerobic, Gram-positive, rod-shaped, and spore-forming bacteria. Spore endurance to unfavorable conditions has assisted the widespread of *Bacillus* [8,9]. Although the culture method is the gold standard for bacteria identification, it is inefficient, time-consuming (more than 1 week), requires laboratory operator expertise, and identification depends on specific microbiological and biochemical testing [7,9,10]. Besides these methods, there are also other detection methods such as polymerase chain reaction (PCR) and enzyme-linked immunosorbent assay (ELISA). However, the PCR-based technique requires sophisticated equipment, complicated techniques, and lengthy processes such as pre-enrichment, DNA extraction, and amplification. To demonstrate pathogenicity, the PCR test on *L. monocytogenes* must be followed by verocytotoxic testing in vitro [11]. ELISA, on the other hand, requires sample enrichment and processing before analysis, has instability of antibodies, and a risk of false positive/negative [12]. An electronic nose (e-nose), on the other hand, has been reported as successful in differentiating different samples according to organic volatile compounds (VOCs) emanated from the samples [13,14]. Recently, e-noses are widely used for analysis in many fields of science and industry (e.g., medicine, safety, the food industry, pharmaceuticals, and the chemical and environmental protection industries) [15]. It has also been reported that e-noses have been applied to identify and classify three bacterial species in different culture media with an accuracy of up to 90% [16]. In addition, the application of commercial and laboratory-made e-noses is also reported as the most prominent example of sensor arrays and pattern recognition systems that measure and compare flavors, odors, and flavors that are easily made and can provide sensitive and selective analysis in real-time [17].

Basically, an e-nose mimics the human nose to differentiate objects according to odor or volatile compounds. It consists of a gas sensor array with global selectivity and chemometric model-based signal analysis [18,19]. The advantages of e-noses compared to existing analytical instruments are that theyare simple in sample preparation (without extraction nor reagent), the e-nose system is easy and inexpensive to operate, and analysis and interpretation of the resultsare also very easy [20,21]. Previously, an e-nose was reported to be able to classify four groups of bacteria of six groups with an accuracy of 94% and 98% when being coupled with a self-organizing map (SOM) and a radial basis function (RBF) network, respectively [22]. Another study reported that an e-nose was able to distinguish *E. coli* and *L. monocytogenes* with an accuracy of 92.4% when using linear discriminant analysis (LDA) [23].

In this study, the lab-made e-nose, comprised of eight MOS gas sensors, was tested for fast detection and differentiation of *L. monocytogenes* and *B. cereus*. To evaluate the performance of the e-nose, chemometric models of LDA, QDA, and SVM were applied for data analysis. Furthermore, 10-fold cross-validation (CV) with 10 repeats was applied to overcome overfitting. A CV is a statistical method where the data were separated into two subsets of training data (as internal validation) and testing data (as external validation). It means that the chemometric models are trained by a subset of training and validated by a testing subset. Before being analyzed using the e-nose, the bacteria were identified with CristieAlkins Munch Peterson (CAMP) and cultured on listeria selective agar (LSA) for *L. monocytogenes*. Meanwhile, *Bacillus cereus* was cultured on *Bacillus cereus* agar (BCA) and mannitol eggyolk polymixin (MYP).

## 2. Materials and Methods

### 2.1. Sample Preparation

The *L. monocytogenes* serotype 4b ATCC 13932 and *B. cereus* ATCC 10876 (purchased from BRIO Food Laboratory, Indonesia) were cultured in tryptic soy broth CM1029 (Oxoid, Hamspire, UK), followed by incubating at 37 °C for 24 h and harvesting by centrifugation at 1600 rpm for 10 min. The recovered pellet was then resuspended in 30% sterile glycerol and kept at −20 °C until further use. The *L. monocytogenes* and *B. cereus* stocks were thawed in a water bath at 37 °C for approximately 2 min or until all ice crystals have melted. Then, the bacterial count was performed by total plate count (TPC) or bacteria counting was carried out prior to determining the total number of the stock culture of each sample, which was used for each replicate, followed by bacteria re-identification. The number of bacteria was calculated in serial dilutions and the growing colonies were enumerated using the Darkfield Quebec colony counter, USA, then cultured at 10^3^–10^4^ CFU/mL in TSB. Subsequently, 3 mL of inoculated TSB was transferred into a sterile falcon tube and covered with a sterile vial plastic bottle. The samples then incubated at 37 °C and e-nose measurement was performed sequentially at 2, 8, 16, 24, 32, 40, and 48 h. For reidentification, the *L. monocytogenes* was Gram stained, followed by being cultured in blood agar, listeria selective agar (LSA), and Christie Atkins Munch Peterson (CAMP). On the other hand, *B. cereus* was Gram and spore stained, followed by being cultured in blood agar, *Bacillus cereus* agar (BCA) and mannitol egg yolk polymixin agar (MYP). The growing colonies from LSA and BCA were then used for biochemical property tests such as catalase, methyl red–voges Proskauer (MR–VP) and carbohydrate utilization. 

### 2.2. Electronic Nose Specification

The lab-made e-nose used in this study consisted of 8 types of MOS gas sensors (from Figaro Inc., Osaka, Japan), as listed in Table 1, namely TGS 813, TGS 822, TGS 823, TGS 826, TGS 2600, TGS 2603, TGS 2612, and TGS 2620. The e-nose was also equipped with an SHT31-D sensor (Sensirion Inc., Tokyo, Japan) for air temperature and humidity monitoring in the inside of the sensor chamber. In addition, the e-nose device comprised of a sampling system, a data acquisition unit (DAQ), and a signal processing framework, as shown in Figure 1. The DAQ for sensor output signal acquisition was built using a microcontroller board, based on the ATmega2560 (or otherwise known as Arduino Mega 2560).

### 2.3. Electronic Nose Measurement and Chemometric Analysis

Figure 2a shows the electrical schematic of the recording signal from the sensor in the e-nose. The MOS-based sensor used in the e-nose requires a power supply for the heater (V_H_) and sensor (V_S_). Before converting to digital format (to ADC), the voltage output from load resistance must be filtered (low-pass filter) and subsequently be amplified. An example of responses of a gas sensor array of the e-nose during delay, sampling, and purging processes is shown in Figure 2b. Prior to using, the e-nose was turned on for 30 min to warm-up the gas sensors. One needle was used to flow air from the environment (as reference gas) and the other was used to flow the gas or volatile compounds from the sample headspace to the sensor chamber. The system was set to automatically sense and record the measured data at a rate of 0.1 s. It means that a dataset of 10 signal values was sent every second from the microcontroller unit to the data logger by RS-232 serial communication in triplicates of 7 repeats. Each measurement cycle in this study was set with a total sampling time of 130 s, consisting of 10 s delay, 60 s sensing and 60 s purging.

The e-nose measurement was performed to the three groups of samples, namely, TSB media (N), *L. monocytogenes* (L) inoculated in TSB, and *B. cereus* (B) inoculated in TSB at 10^3^–10^4^ CFU/mL after incubation at 37 °C for 2, 8, 16, 24, 32, 40, and 48 h. After incubation, the samples were set on a hot plate at 47 to 53 °C (setpoint heater temperature) during the sampling process to promote the release of volatile compounds. Each group consisted of three samples for each incubation time, repeated 7 times at various days of sampling.

The total data of measurement are 336 samples, consisting of 7 incubation times × 6 independent repeats = 42 data of TBS blank (Neutral (N)), 7 incubation times × 21 independent repeats = 147 data of *L. monocytogenes*(L), and 7 incubation times × 21 independent repeats = 147 data of and *B. cereus* (B). In this case, the sample of TBS blank was only repeated six times because of almost the same reading of each measurement. The independent repeat means the different culture stocks. Each sample was measured by eight types of gas sensors, one temperature sensor, and one humidity sensor so that the sensor responses contained 10 sensors × 1301 data lines. 

Prior to an analysis by chemometric models, the pre-processing by fast Fourier transform (FFT) and subsequently the scaling by robust scaler were applied to raw data to extract the features and to scale data into an interquartile range. FFT is often used to convert the signal from the time domain to the frequency domain. Since the e-nose measurement setting for each cycle of all experiments is the same, applying FFT results in amplitude or the maximum value of each sensor response. Meanwhile, the scaling by robust scaler was used to shrink the range such that the range is interquartile so that it is robust to outliers. The chemometric models used in this analysis included linear and quadratic discriminant analyses (LDA and QDA) and support vector machine (SVM) for pattern recognition and classification. The performance of the models was then compared to obtain the highest accuracy.

## 3. Results

### 3.1. Bacteria re-Identification Colony Counting

Prior to utilization as a sample for testing, the bacteria colony must be identified and counted to make sure of similar initial conditions. Gram and spore stainings of *L. monocytogenes* and *B. cereus* are needed for this purpose. Figure 3 shows the photographs of *L. monocytogenes* that grew on LSA media and CAMP tests that show hemolysis. This also shows that *B. cereus* grew well in BCA and MYP media. The number of bacteria before inoculation was confirmed by the total number of plates (TPC; 10^3^–10^4^ CFU/mL). Table 2 shows Gram and spore stainings, and biochemical tests of *L. monocytogenes* and *B. cereus.*

### 3.2. Bacterial Classification

In this study, the data matrix of 336 samples in 21 subgroups was analyzed by 8 gas sensors. A radar plot, as shown in Figure 4, is used for illustrating the variability between average sensor response profiles of each sample group of the bacterial samples evaluated. The radius of the plot has been normalized. These variabilities could be attributed to the different MVOC composition of the bacterial samples.

To analyze the performance of the e-nose, data from all measurements were processed with the LDA, QDA, and SVM chemometric models. The total data after applying pre-processing with FFT was the same as the total number of samples (336). This consisted of 147 data B, 42 data N, and 147 data L. During the training step, a repetitive 10-fold cross-validation (CV) procedure, and 10 repetitions were carried out for data analysis. For this purpose, 70% of each group’s datawas randomly divided for training data and for internal validation, while the remaining of 30% was used for external validation.

Figure 5 shows the LD1 plot of LDA of three groups of N with B, N with L, and B with L. Another way to visualize the correctness or the accuracy of this e-nose is by using the confusion matrix, as shown in Figure 6. It explains the relation between the actual measurement and the predicted value. The diagonal of the confusion matrix shows the correct data (true negative (TN) and true positive (TP)), while the others are misclassified data (false negative (FN) and false positive (FP)).For example, in Figure 6a, the data N were correct and miss-classified as 30 and 12, respectively. Meanwhile, data B was correct and miss-classified as 145 and 2, respectively. The others can be interpreted with the same method. The accuracy can be calculated by (TP+TN)/total. 

Table 3 explains the possible relationship between MVOCs (obtained from GC–MS analysis based on existing references) with related functional groups and types of sensors that respond to those volatile compounds (active sensors). Thus, each sample is characterized by a unique pattern to be distinguished from the other samples.

## 4. Discussion

As shown in Table 2, the Gram stain of *L. monocytogenes* indicated a Gram-positive, rod-shaped bacteria. The colonies grown in blood agar showed beta-hemolysis. As shown in Figure 3a, the hemolysis zone in Listeria was formed by synergism with *S.aureus* or was not formed by synergism with *Rhodococcusequi*. The 58-kDa (LLO) protein released by *L. monocytogenes* is highly hemolytic to sheep erythrocytes when combining with supernatants of *S. aureus* but not with *R. equi*. The CAMP test showed the presence of hemolytic β, in reference to *R. equi* [25], hemolytic β with *S. aureus* and LLO shows that the bacterium is *L. monocytogenes*. The re-identification of *L. monocytogenes* was supported by biochemical test results, as shown in Table 2, where carbohydrate utilization showed fermentation of rhamnose, glucose, maltose, non-fermented xylose, and mannitol, which match the characteristics of *L. monocytogenes* [26].

The re-identification of *B. cereus*, on the other hand, was shown in Table 2, where Gram staining shows that the cells are purple androd-shaped or coccobacillus, whereas in spore staining, green spores appear in the terminal or subterminal, which is in accordance with [26]. The re-identification is also proven by the growth of a typical peacock blue fimbriate colony (3–5 mm) surrounded by a blue zone of egg yolk hydrolysis against a green/green-yellow background on the *Bacillus cereus* agar medium in Figure 3d. The biochemical test in Table 2 also shows the same results as previous studies [18,19].

As shown in Figure 4, the voltage sensor response of the MVOCs during the incubation period showed a distinct pattern between the three groups. The intensity of the voltage sensor response is important in distinguishing between samples. The sensor arrays were sensitive, fast, and potentially used for MVOC characterization although each sensor has a varied level of sensitivity and limit for detection of each bacterial profile [21].

Figure 5 shows the LD1 plots of LDA of three groups of bacterial samples. According to LD1, the e-nose coupled with LD1 shows accuracies of 93%, 94%, and 52% in differentiating N from B, N from L, and L from B, respectively (as shown in Figure 6a–c). Especially for differentiating L from B, a lot of data were overlapped. Therefore, LDA is not suitable for obtaining a high accuracy of the e-nose. This motivated us to apply non-linear chemometric models of QDA and SVM for analyzing the same data of the e-nose. Compared to LDA, the e-nose with quadratic discriminant analysis (QDA) shows higher accuracies(Figure 6d–f) of 98%, 96%, and 56% in differentiating N from B, N from L, and L from B, respectively. Meanwhile, as shown in Figure 6g–i, the e-nose with support vector machine (SVM) analysis shows high accuracies of 98% in differentiating both L from N and B from N, and 83% in differentiating L from B. It indicates that the e-nose combined with SVM shows the highest performance in detecting *L. monocytogenes* and *B. cereus.* As listed in Table 1 and Table 3, each type of bacteria produces marker gases that useful for differentiation. The MVOCs produced by *L. monocytogenes* when incubated in TSB medium included 2-undecanone, 2-nonanone/1-undecene, dimethyl trisulfide, aldehydes, acetone, ketone, 3-methyl-butanal, 2-methyl-butane, and 3-hydroxy- 2-butanone, which can be used as a marker of the growth phase [25,27]. Meanwhile, *B. cereus* produces 2-undecanone and dimethylsulfide, 4-hydroxy-2-butanone, ethyl acetate, n-pentanal, n-hexanal, as well as octanal, and the main yield is pentadecanal [28,29]. In addition, *L. monocytogenes* and *B. cereus* produce the same MVOCs, namely, 2-undecanone [29]. Trypticsoy broth (TSB) media produces methanethiol, aldehydes, and acetone and the main gas marker is 2-methyl-propanol [30]. The MVOCs produced by each group indicate that no similarities of a marker of MVOC types are found between group N and group B. 

In general, bacterial growth is divided into four distinct phases, namely, the adaptation phase (lag phase), the growth phase (exponential phase), the balance phase (stationary phase), and the death phase [31]. The lag or adaptation phase for *B. cereus* is 9.01 h [32], while for *L. monocytogenes*, the initial growth or adaptation phase occurs at the incubation period of 0 to 6 h 219]. The lag phase the bacterial adaptation occurs, where the bacteria do not reproduce immediately and the number of cells remain constant; cells are metabolically active and only increase in cell size. Then entering the log phase, the number of cells increases logarithmically, and each cell generation occurs at the same time interval as the previous one. The log phase continues until nutrients run out or toxic products accumulate, at which time the cell growth rate slows and some cells may start to die. The log phase of bacterial growth is followed by the stationary phase; the size of the bacterial population remains constant although some cells continue to divide and others begin the die-phase [21,31,33].

## 5. Conclusions

This study demonstrates the perspective of using a lab-made e-nose with MOS gas sensors coupled with chemometric models to correctly classify the existence of *L. monocytogenes* and *B. cereus* on TSB media. Here, the e-nose coupled with three chemometric models (LDA, QDA, SVM) was evaluated to detect *L. monocytogenes* and *B. cereus* on TSB media. It is found that e-nose/SVM results in the highest performance, with an accuracy of 98%. Thus, the proposed tool allows a preliminary, fast, and cost-effective and accurate classification model that can be used to identify the presence of *L. monocytogenes* and *B. cereus* in routine and fast assessments of food products in animal quarantine.

## Figures and Tables

**Figure 1 vetsci-07-00020-f001:**
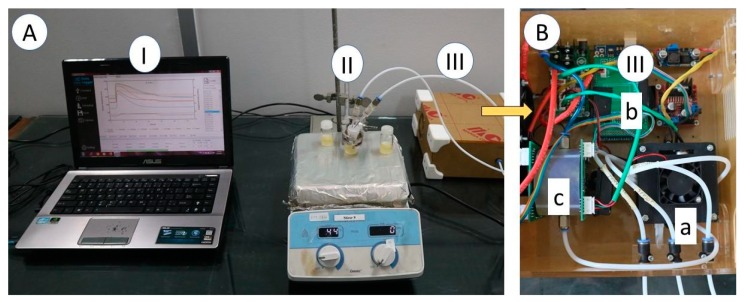
(**A**) E-nose measurement set-up. I: a personal computer with a software of DAQ and chemometric models, II: sample on the hot plate, and III: the main part of e-nose; (**B**) III: main part of electronic nose device. a: sampling system, b: DAQ and controller, c: sensor chamber.

**Figure 2 vetsci-07-00020-f002:**
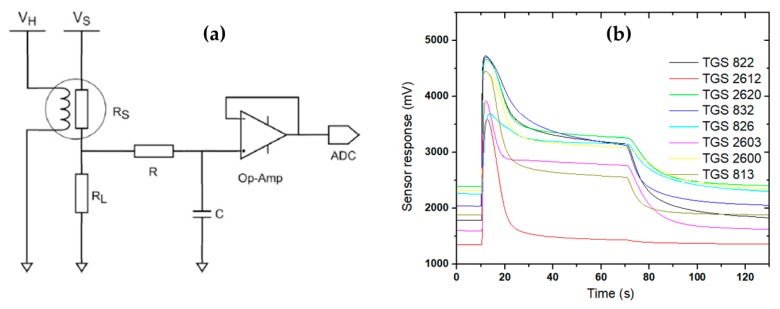
(**a**) Electrical schematic of recording a signal from a sensor in this e-nose. V_H_ and V_S_ are voltage sources for heater and sensor, respectively; RS and R_L_ are sensor resistance and load resistance, respectively; R and C for the low-pass filter. (**b**) Typical of gas sensor response of e-nose during the delay, sampling and purging processes.

**Figure 3 vetsci-07-00020-f003:**
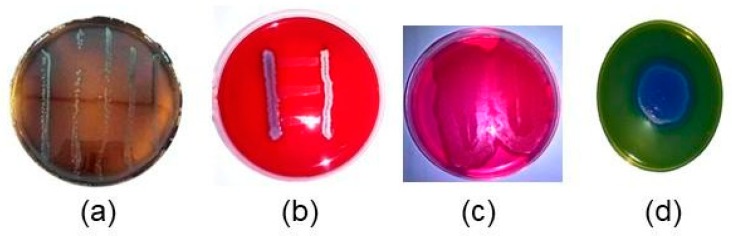
(**a**) *Listeria monocytogenes* on listeria selective agar (LSA) media (**b**) Christie Atkins Munch Peterson (CAMP) test result; *Bacillus cereus* colonies (**c**) on mannitol egg yolk polymixin agar (MYP) media (**d**) on BCA media.

**Figure 4 vetsci-07-00020-f004:**
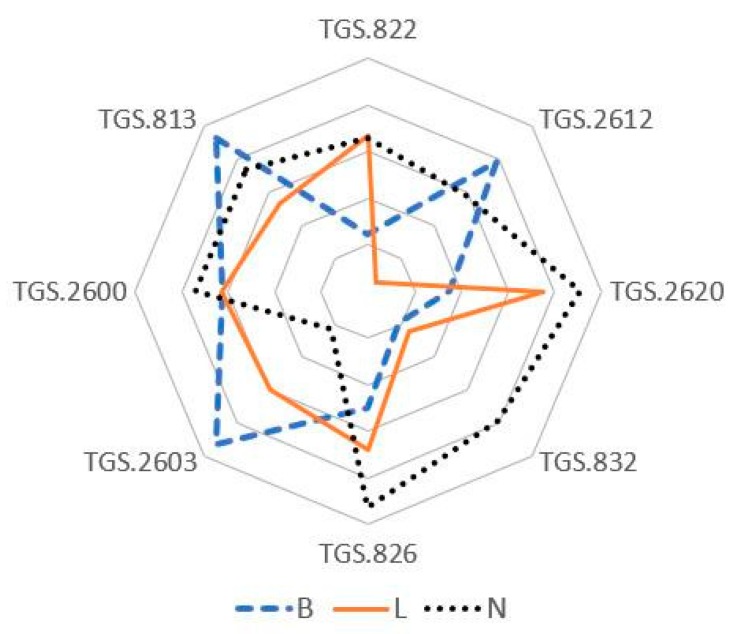
Radar plot of the average sensor responses obtained with the gas sensor array for each bacterial sample. N: TSB uninoculated; L: TSB inoculated with *L. monocytogenes*, B: TSB inoculated with *B. cereus*.

**Figure 5 vetsci-07-00020-f005:**
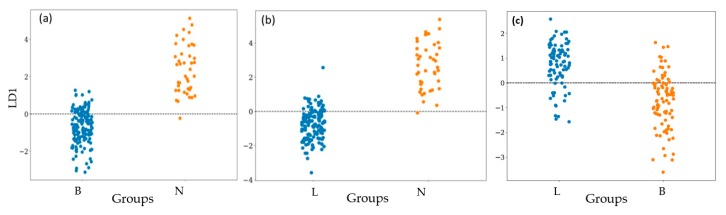
Classification of group N, B, and L after incubation for 48 h using linear discriminant analysis1 (LDA1). (**a**): N vs. B and (**b**): N vs. L, and (**c**) B vs. L.

**Figure 6 vetsci-07-00020-f006:**
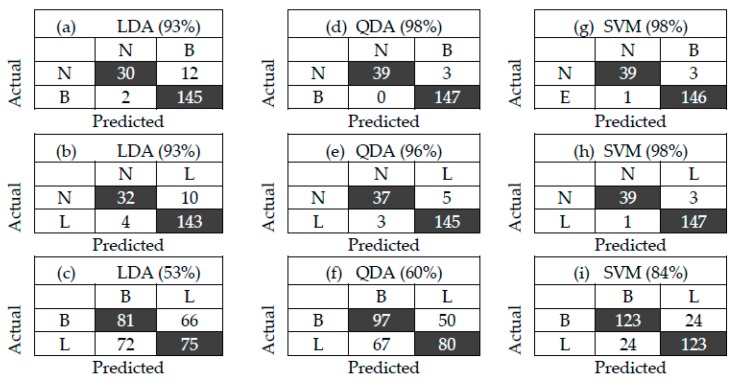
Confusion matrix of two groups with different chemometric models of LDA (**a**–**c**), quadratic discriminant analysis (QDA) (**d**–**f**), and support vector machine (SVM) (**g**–**i**). The number in parenthesis indicates the accuracy.

**Table 1 vetsci-07-00020-t001:** Sensor types used in the lab-made e-nose and the targeted volatile compounds [24].

Type of Sensor	Targeted Volatile Compounds
TGS813	Methane, ethanol, propane, isobutane, hydrogen, and carbon monoxide
TGS822	Ethanol, acetone, benzene, n-hexane, isobutane, carbon monoxide, and methane
TGS823	Combustible gas, i.e., ethanol
TGS826	Ammonia
TGS2600	Methane, carbon monoxide, isobutane, ethanol, and hydrogen
TGS2603	Hydrogen, H_2_S, ethanol, methyl mercaptan, and trimethylamine
TGS2612	Ethanol, methane, isobutane, and propane
TGS2620	Methane, carbon monoxide, isobutane, hydrogen, and ethanol

**Table 2 vetsci-07-00020-t002:** Gram and spore staining, and biochemical tests of *L. monocytogenes* and *B. cereus.*

Test	*L. monocytogenes*	*B. cereus*
Catalase	+	+
*Sulfur Indole Motility* (SIM)	+	+
CAMP	+	na
Voges–Proskouer	+	+
Carbohydrate		
Glucose	+	+
Maltose	na	+
Mannitol	−	−
Sucrose	+	+
Rhamnose	+	na
Xylose	−	na
Gram staining	+	+
Spore staining	na	+

+: positive,−: negative, na: not available.

**Table 3 vetsci-07-00020-t003:** Metabolic volatile organic compounds (MVOCs) and possible active sensor.

MVOCs	Functional Groups	Active Sensors
***Listeria monocytogenes***		
2-undecanone	methyl vinyl ketone	TGS 822
2-nonanone/1-undecene	acrylic alkanes	TGS 813, TGS 2612, TGS 2620, TGS 2600
dimethyl trisulfide	2,3,4-trithiapentane	TGS 813, TGS 2612, TGS 2620, TGS 2600
aldehydes	aldehyde	TGS 823, TGS 2600, TGS 2603
ketones	ketones	TGS 822
3-methyl-butanal	butanal	TGS 823, TGS 2600, TGS 2603
acetone	propanone	TGS 813, TGS 2612, TGS 2620, TGS 2600
2-methyl-butane	isopentane	TGS 813, TGS 2612, TGS 2620, TGS 2600
3-hydroxy-2-butanone	methyl acetoin	TGS 813, TGS 2612, TGS 2620, TGS 2600
***Bacillus cereus***		
2-undecanone	metal ketones	TGS 822
dimethylsulfide	2,3,4-trithiapentane	TGS 813, TGS 2612, TGS 2620, TGS 2600
4-hydroxy-2-butanone	methyl acetone	TGS 822
ethyl acetate	ester	TGS 813, TGS 2612, TGS 2620, TGS 2600

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
