# Peer review of "Lab-Made Electronic Nose for Fast Detection of Listeria monocytogenes and Bacillus cereus"

_vetsci, 2020, doi:10.3390/vetsci7010020_

Round 1

Reviewer 1 Report

The abstract and methodology should be changed as the explanation is not clear. The confusion matrix also should be explained better. Table 2 is not a table but a figure. So, would be good to make separate tables instead to the figure. Describe the tables 3 and 4 better since these are very difficult to understand. A table must be placed with the reported MVOCs and match them with the sensor responses. I recommend to make a representation of the logarithmic growth of the bacteria. Review the number of the graphs and tables and relate them in the text when it is the case to explain them

Author Response

COMMENTS AND SUGGESTIONS FOR AUTHORS:

Reviewer #1:

The abstract and methodology should be changed as the explanation is not clear. The confusion matrix also should be explained better. Table 2 is not a table but a figure. So, would be good to make separate tables instead to the figure. Describe the tables 3 and 4 better since these are very difficult to understand. A table must be placed with the reported MVOCs and match them with the sensor responses. I recommend to make a representation of the logarithmic growth of the bacteria. Review the number of the graphs and tables and relate them in the text when it is the case to explain them.

Authors answer: We acknowledge the overall positive and kind remarks of the reviewer.

Detailed remarks about the paper are as follows:

The abstract and methodology should be changed as the explanation is not clear

Authors response: We have revised the abstract to make sure including objectives, backgrounds, material and methods, results, and discussion. We have also revised the methodology to make clearer the explanation (Sample Preparation, Electronic Nose Specification, Electronic Nose Measurement and Chemometric Analysis).

The confusion matrix also should be explained better

Authors response: The explanation about the confusion matrix (Figure 6) have been added.

In-text line 210: “Another way to visualize the correctness or the accuracy of this e-nose is by using the confusion matrix, as shown in Figure 6. It explains the relation between the actual measurement and the predicted value. The diagonal of the confusion matrix shows the correct data (true negative (TN) and true positive (TP)), while the others are misclassified data (false negative (FN) and false positive (FP)). For example, in Figure 6(a), the data N were correct and miss-classified as 30 and 12, respectively. Meanwhile, data B was correct and miss-classified as 145 and 2, respectively. The others can be interpreted with the same method. The accuracy can be calculated by (TP+TN)/total.”

In-text line 251: “Figure 5 shows the LD1 plots of LDA of three groups of bacterial samples. According to LD1, the e-nose coupled with LD1 shows the accuracy of 93%, 94%, and 52% in differentiating N from B, N from L, and L from B, respectively (as shown in Figure 6 (a-c)). Especially for differentiating L from B, a lot of data are overlapping. Therefore, LDA is not suitable for obtaining a high accuracy of the e-nose. This motivates us to apply non-linear chemometric models of QDA and SVM for analyzing the same data of e-nose. Compared to LDA, e-nose with quadratic discriminant analysis (QDA) shows a higher accuracy (Figure 6 (d-f)), of 98%, 96%, and 56% in differentiating N from B, N from L, and L from B, respectively. Meanwhile, as shown in Figure 6 (g-i), e-nose with support vector machine (SVM) analysis shows high accuracy of 98% in differentiating both L from N, and B from N, and 83% in differentiating L from B. It indicates that the e-nose combined with SVM shows the highest performance in detecting L. monocytogenes and B. cereus.

Table 2 is not a table but a figure. So, would be good to make separate tables instead to the figure.

Authors response: Table 2 number 2 has been revised to be Figure 6, and its figure caption has also been added.

In-text line 219: “Figure 6. Confusion matrix of two groups with different chemometric models of LDA (a, b, c), QDA (d, e, f), and SVM (g, h, i). The number in parenthesis indicates the accuracy.”

Describe the tables 3 and 4 better since these are very difficult to understand. A table must be placed with the reported MVOCs and match them with the sensor responses.

Authors response: Tables 3 and 4 have been deleted and the accuracies of each chemometric model in Table 3 have been inserted in the parentheses in Figure 6. Table 4 was deleted because of the irrelevant with the current study. 

I recommend to make a representation of the logarithmic growth of the bacteria.

Authors response: We did not count the number of bacteria in each incubation period, but we only count the number of bacteria by the Total Plate Count (TPC) method at the beginning of each repetition. This is to ensure that e-nose measurements are carried out on samples with nearly the same number of colonies and conditions.

Review the number of the graphs and tables and relate them in the text when it is the case to explain them.

Authors response: We reviewed and explained the graphs and tables and their relation in Discussion section.

Reviewer 2 Report

Dear authors,

congratulations for your research.

I suggest some minor revisions:

line 65 insert more references. line 67 the sample preparation is a delicate aspect also for the e-nose analysis. I think that you can modify the expression "withouth complicate..." by saying that e-nose allows to perform rapid, cheap and continuous measurement. line 218 insert dot at the end.

I think that you should improve results description and discussion. I suggest to use only one section in order to provide to readers considerations and comments for each result presented, thereby guiding their understanding of each plot and table. 

Author Response

We acknowledge the overall positive and kind remarks of the reviewer.

Detailed remarks about the paper are as follows:

line 65 insert more references.

Authors response: We added more relevant references

In-text line 64: “To demonstrate pathogenicity, the PCR test on L. monocytogenes must be followed by verocytotoxic testing in vitro [11]. ELISA, on the other hand, requires sample enrichment and processing before analysis, instability of antibodies, and risk to false positive / negative [12]. An electronic nose (e-nose), on the other hand, has been reported as successful in differentiating different samples according to organic volatile compounds (VOCs) emanated from the samples [13,14]. Recently, e-nose is widely used for analysis in many fields of science and industry (e.g., medicine, safety, the food industry, pharmaceuticals, and the chemical and environmental protection industries) [15]. It has also been reported that e-nose has been applied to identify and classify three bacterial species in different culture media with an accuracy of up to 90% [16]. In addition, the application of commercial and laboratory-made e-noses is also reported as the most prominent example of sensor arrays and pattern recognition systems that measure and compare flavors, odors, and flavors that are easily made and can provide sensitive and selective. analysis in real-time [17].”

line 67 the sample preparation is a delicate aspect also for the e-nose analysis.

Authors response: We emphasize that the advantages of the e-nose analysis compared to existing analytical instruments.

I think that you can modify the expression "without complicate..." by saying that e-nose allows to perform rapid, cheap and continuous measurement.

Authors response: We changed the expression to highlight the advantages of an e-nose.

In-text line 79: “The advantages of e-nose compared to existing analytical instruments are that it is simple in sample preparation (without extraction nor reagent), easy and inexpensive in operating the e-nose system, and also very easy in analysis and interpretation of the results [20,21].”

line 218 insert dot at the end.

Authors response: We added dot at the end of in-text line 238.

I think that you should improve results description and discussion.

Authors response: As suggested, we have improved the whole description and discussion of results.

I suggest to use only one section in order to provide to readers considerations and comments for each result presented, thereby guiding their understanding of each plot and table.

Authors response: As suggested, we made the results separated from the discussion by following the journal template.

Reviewer 3 Report

The paper presents posiibility of use the e-nose with LDA, QDA and SVM methods for fast detection of pathogenic bacteria: Listeria monocytogenes and Bacillus cereus.

Reviewer asks authors to response the following comments:

There are two Tables 1 and Tables 2 in the manuscript.

Line 32: In the Abstract there is no informations about what the symbol N means?

Line 65: Two references are not enough. Sensors are increasingly used to detect and distinguish microorganisms. Please see:

doi: 10.1016/j.measurement.2017.11.029 doi: 10.1016/j.ijfoodmicro.2018.06.020 doi: 10.1109/IECBES.2014.7047589

Line 127: Arduino MEGA isn't a microcontroller and doesn't have 16-bit ADC converters

Figure 2: Please add to this figure electrical schematic of recording signals from sensors

Table 1(2): it is not necessary to explain sample names, since they are not presented anywhere else on the charts/plots/schematics

Line 151: Please explain why FFT was used?

Figure 4: Presented radar plot shows a lot of informations about sensors response. Please try to link this informations with specific compounds presented in Lines 229-234 (using Table 1(1)).

Lines 188 and 190: what (iii) means in the Figure reference?

Figures 5 and 6 - please make one Figure with labels (a) (b) and (c)

Table 3 - it is not necessary to present another table since this information results from the Table 2(2) - please add this informations to Tabel 2(2)

Table 4 - no data for SVM

Table 3 and 4 - there is no reference in the text to this tables

The article must be completely edited to allow it to continue in the evaluation.

Author Response

We acknowledge the overall positive and kind remarks of the reviewer.

Detailed remarks about the paper are as follows:

There are two Tables 1 and Tables 2 in the manuscript.

Authors response: We have made correction; the second Table 2 was changed to be Figure 6.

Line 32: In the Abstract there is no information about what the symbol N means?

Authors response: We deleted symbol N in the abstract to avoid misunderstanding.

Line 65: Two references are not enough. Sensors are increasingly used to detect and distinguish microorganisms. Please see: doi: 10.1016/j.measurement.2017.11.029; doi: 10.1016/j.ijfoodmicro.2018.06.020; doi: 10.1109/IECBES.2014.7047589..

Authors response: We added three relevant references including the above-suggested references.

In-text line 69: “Recently, e-nose is widely used for analysis in many fields of science and industry (e.g., medicine, safety, the food industry, pharmaceuticals, and the chemical and environmental protection industries) [15]. It has also been reported that e-nose has been applied to identify and classify three bacterial species in different culture media with an accuracy of up to 90% [16]. In addition, the application of commercial and laboratory-made e-noses is also reported as the most prominent example of sensor arrays and pattern recognition systems that measure and compare flavors, odors, and flavors that are easily made and can provide sensitive and selective. analysis in real-time [17].

Line 127: Arduino MEGA isn't a microcontroller and doesn't have 16-bit ADC converters.

Authors response: We revised according to the datasheet of the Arduino MEGA.

In-text line 124: “The DAQ for sensors output signals acquisition was built using a microcontroller board, based on the ATmega2560 or known as Arduino Mega 2560.”

Figure 2: Please add to this figure electrical schematic of recording signals from sensors.

Authors response: We added an electrical schematic of recording signals from sensors and its description.

In-text line 133: “Figure 2(a) shows the electrical schematic of the recording signal from the sensor in the e-nose. The MOS-based sensor used in the e-nose requires a power supply for the heater (VH) and for sensor (VS). Before converting to digital format (to ADC), the voltage output from load resistance must be filtered (low-pass filter) and subsequently be amplified. An example of responses of a gas sensor array of the e-nose during delay, sampling and purging processes is shown in Figure 2(b). Prior to using, the e-nose was turned on for 30 minutes for warming-up the gas sensors. One needle was used to flow air from the environment (as reference gas) and the other was used to flow the gas or volatile compounds from the sample headspace to the sensor chamber. The system was set to automatically sense and record the measured data at a rate of 0.1 s. It means that a dataset of 10 signal values was sent every second from the microcontroller unit to the data logger by RS-232 serial communication in triplicates of 7 repeats. Each measurement cycle in this study was set with a total sampling time of 130 s consisted of 10 s delay, 60 s sensing and 60 s purging.”

In-text line 146: “Figure 2. (a) Electrical schematic of recording a signal from a sensor in this e-nose. VH and VS are voltage sources for heater and sensor, respectively; RS and RL are sensor resistance and load resistance, respectively; R and C for low-pass filter. (b) Typical of gas sensor response of e-nose during the delay, sampling and purging processes.”

Table 1(2): it is not necessary to explain sample names, since they are not presented anywhere else on the charts/plots/schematics.

Authors response: We deleted Table 1 (2) because of not necessary.

Line 151: Please explain why FFT was used?

Authors response: We added explanation about FFT 

In-text line 163: “Prior to an analysis by chemometric models, the pre-processing by Fast Fourier Transform (FFT) and subsequently the scaling by robust scaler were applied to raw data to extract the features and to scale data into an interquartile range. FFT was often used to convert the signal from the time domain to the frequency domain. Since the e-nose measurement setting for each cycle of all experiments is the same, applying FFT results in amplitude or maximum value of each sensor response. Meanwhile, the scaling by robust scaler was used to shrink the range such that the range is interquartile so that it is robust to outliers. The chemometric models used in this analysis included linear and quadratic discriminant analysis (LDA and QDA) and support vector machine (SVM) for pattern recognition and classification. The performance of them was then compared to obtain the highest accuracy.”

Please try to link this informations with specific compounds presented in Lines 229-234 (using Table 1(1)).

Authors response: We made a correlation to link with specific compounds presented in the Discussion section with the active sensors. Here, the active sensor means the sensor that possible responses to related MVOCs and functional groups.

In-text line 222: “Table 3. Metabolic volatile organic compounds (MVOCs) and possible active sensor”

MVOCs

Functional Groups

Active sensors

Listeria monocytogenes

2-undecanone

methyl vinyl ketone

TGS 822

2-nonanone/1-undecene

acrylic alkanes

TGS 813, TGS 2612, TGS 2620, TGS 2600

dimethyl trisulfide

2,3,4-trithiapentane

TGS 813, TGS 2612, TGS 2620, TGS 2600

aldehydes

aldehyde

TGS 823, TGS 2600, TGS 2603

ketones

ketones

TGS 822

3-Methyl-butanal

butanal

TGS 823, TGS 2600, TGS 2603

Acetone

propanone

TGS 813, TGS 2612, TGS 2620, TGS 2600

2-Methyl-butane

isopentane

TGS 813, TGS 2612, TGS 2620, TGS 2600

3-hydroxy- 2-butanone

methyl acetoin

TGS 813, TGS 2612, TGS 2620, TGS 2600

Bacillus cereus

2-undecanone

metal ketones

TGS 822

dimethylsulfide

2,3,4-trithiapentane

TGS 813, TGS 2612, TGS 2620, TGS 2600

4-hydroxy-2-butanone

methyl acetone

TGS 822

ethyl acetate

ester

TGS 813, TGS 2612, TGS 2620, TGS 2600

Lines 188 and 190: what (iii) means in the Figure reference?

Authors response: We revised the description in the Discussion section.

In-text line 251: “Figure 5 shows the LD1 plots of LDA of group N with B, group N with L, and group B with L. According to LD1, the e-nose coupled with LD1 shows accuracy of 93%, 94%, and 52% in differentiating N from B, N from L, and L from B, respectively (as shown in Figure 6 (a-c)). Especially for differentiating L from B, a lot of data are overlapping. Therefore, LDA is not suitable for obtaining a high accuracy of the e-nose. This motivates us to apply non-linear chemometric models of QDA and SVM for analyzing the same data of e-nose. Compared to LDA, e-nose with quadratic discriminant analysis (QDA) shows a higher accuracy (Figure 6 (d-f)), of 98%, 96%, and 56% in differentiating N from B, N from L, and L from B, respectively. Meanwhile, as shown in Figure 6 (g-i), e-nose with support vector machine (SVM) analysis shows high accuracy of 98% in differentiating both L from N, and B from N, and 83% in differentiating L from B. It indicates that the e-nose combined with SVM shows the highest performance in detecting L. monocytogenes and B. cereus.

Figures 5 and 6 - please make one Figure with labels (a) (b) and (c)

Authors response: We combined Figures 5 and 6 to be Figure 5 with labels (a) (b) and (c).

Table 3 - it is not necessary to present another table since this information results from the Table 2(2) - please add this informations to Tabel 2(2)

Authors response: We deleted Tables 3 and 4. Information about accuracy is added to the confusion matrix. Table 2(2) is changed to be in Figure 6. Table 3 after revision is about the possible relationship between metabolic volatile organic compounds (MVOCs) and active sensor.

Figure 4: Presented radar plot shows a lot of information about sensors response.

Authors response: We revised the explanation of Figure 4.

In-text line 189: “A radar plot, as shown in Figure 4, was used for illustrating the variability between average sensor response profiles of each sample group of the bacterial samples evaluated. The radius of the plot has been normalized. These variabilities could be attributed to the different MVOC composition of the bacterial samples.”

Table 4 - no data for SVM, Table 3 and 4 - there is no reference in the text to this tables.

Authors response: We deleted Table 4 because it is not necessary.

Round 2

Reviewer 3 Report

All corrections have been inserted in the manuscript.